# Assessment of knowledge, attitudes, and practices among community pharmacists in Lahore regarding antibiotic dispensing without prescription: A cross-sectional study

Muhammad Nabeel [1,2‡]*, Khubaib Ali[1,2‡], Muhammad Rehan Sarwar[2], Imran Waheed[3]

**1** Department of Oncology Pharmacy, Cancer Care Hospital & Research Centre, Lahore, Punjab, Pakistan, **2** Department of Clinical Pharmacy, Akhtar Saeed College of Pharmaceutical Sciences (ACPS), Lahore, Punjab, Pakistan, **3** Department of Pharmacognosy, Akhtar Saeed College of Pharmacy, Canal Campus (ASCP), Lahore, Punjab, Pakistan

‡ MN and KA are contributed equally to this work and shared the first authorship.
* nabeelsheikh26@gmail.com

**Data Availability Statement:** All relevant data are within the manuscript.

## Abstract

### Objectives

The irrational dispensing practices are responsible for antibiotic abuse and the spread of antibiotic resistance. Thus, the present study aims to evaluate the knowledge, attitudes, and practices of community pharmacists (CPs) regarding dispensing antibiotics without prescription (DAwP).

### Method

A descriptive, cross-sectional study was conducted between March 1, 2023, and March 31, 2023, in community pharmacy settings of Lahore, Pakistan. A self-administered and pre-tested questionnaire was used for the data collection. Logistic regression analysis was used to determine the factors associated with the practices of community pharmacists. Data were analyzed using SPSS (version 26) and MS Office (2016).

### Results

Among 359 respondents, many strongly agreed/agreed with the statements *"DAwP is contributing to the development of antimicrobial resistance"* (83%, n = 298) and *"Antibiotic resistance has become a public health issue"* (81.9%, n = 249). Overall, most of the community pharmacists claimed that the unwillingness of patients to visit physicians for non-serious infections (75.2%, n = 270) and good knowledge of pharmacists about the use of antibiotics (51%, n = 183) were the most common reasons attributable to dispensing of antibiotics without prescription. Cephalosporin (n = 260, 72.4%), penicillin (n = 254, 70.8%), and tetracyclines (n = 170, 47.4%) were the most commonly dispensed classes of antibiotics without prescription due to cold, flu and diarrhea. Most community pharmacists never/sometimes warn patients about the potential side effects of medicines (79.1%, n = 284). Logistic

**Funding:** The author(s) received no specific funding for this work.

**Competing interests:** The authors have declared that no competing interests exist.

regression analysis revealed that community pharmacists 31–40 years of age (OR = 0.568, 95%CI = 0.348–0.927, *p-value* = 0.024) were significantly less associated with poor practices of dispensing antibiotics without prescription (DAwP) while those who were 'Managers' (OR = 4.222, 95%CI = 2.542–7.011, *p-value* = <0.001), had 3–5 years of experience (OR = 2.241, 95%CI = 1.183–4.243, *p-value* = 0.013), dispensed ≤25 antibiotics per day (OR = 12.375, 95%CI = 5.177–29.583, *p-value* = <0.001), were more likely to be associated with poor practices of dispensing of antibiotics without prescription.

## Conclusion

The community pharmacists had adequate knowledge, positive attitudes, and poor practices towards DAwP. Demographical factors such as age, job status, and work experience were the determinants of community pharmacists' practices towards dispensing antibiotics without prescription (DAwP). Hence, a multifaceted approach, including educational interventions, is needed to reduce the dispensing of antibiotics without prescription (DAwP).

## Introduction

The significant rise in the prevalence of infectious diseases has threatened public health and economically burdened healthcare systems all over the globe. The World Health Organization (WHO) has ranked these illnesses among the top ten leading causes of mortalities [1–3]. Although the advent of antibiotics declined the mortality and morbidity rates associated with infectious diseases, developing resistance against single or multiple antibiotics by microbes has posed a new challenge for healthcare professionals [4, 5]. According to data from the Centers for Disease Control and Prevention (CDC), the United States witnesses approximately 2 million new instances of antibiotic-resistant bacterial infections annually. These infections not only pose a significant health challenge but also result in the tragic loss of around 23,000 lives each year due to related complications [6, 7]. In the comparison of developed countries, the unavailability of a national action plan has made this issue more life-threatening for many low and middle-income countries (LMICs). The WHO has estimated that only 25% of developing nations have a national action plan to overcome this dreadful condition [8]. The misuse and irrational use of antibiotics may give suboptimal therapeutic outcomes and contribute towards the development of resistance [9].

Dispensing is of crucial importance in the medication processing system. The WHO also recommends rational dispensing practices for optimal therapy and patient safety [10]. Dispensing antibiotics without prescription (DAwP) can be categorized as irrational and can act as a driving source in the spread of antibiotic-resistant microbial strains. Although DAwP is considered illegal in many community pharmacy settings of high, middle, and low-income countries, antibiotics can be made available as over-the-counter (OTC) drugs. Evidence suggests that nearly half of the sale of antibiotics is attributable to this malpractice [11]. European Union (EU) also forbids DAwP. Still, this practice is common in many of its member states, e.g., Spain, where inadequate communication skills of pharmacists, acquiescence on patient demand, and lack of continuing medical education (CME) are the major contributing factors [12]. A multicenter cross-sectional study conducted in China revealed that 55.9% of the antibiotics were dispensed to pediatric patients while 77.7% were sold to adult patients without medical prescription [13]. Studies reveal that irrational dispensing practices are frequently

observed in Middle Eastern countries like Saudi Arabia [14, 15], Egypt [16], Jordan [17], and Syria [18]. Therefore, in 2017, the WHO updated the essential medicines list (EML) in which antibiotics are categorized as "access group," "watch group," and "reserve group" to limit the excessive and irrational use of these agents [19].

In Pakistan, the Drug Regulatory Authority (DRAP), which works under the federal government, regulates the sale, purchase, and rational use of medicines. Still, the trend of making all medicines available, especially antibiotics, without prescription is very high [20]. Financial limitations often prompt patients to resort to self-medication, seek advice from unqualified practitioners, or borrow medications from neighbours instead of consulting healthcare professionals. This trend is exacerbated by the lack of well-established antimicrobial stewardship (AMS) programs, which are crucial for guiding appropriate medication use [21]. Data regarding dispensing practices in Pakistan is unavailable, but the previously published literature revealed that in this region, most antibiotics are readily available and dispensed without prescription [22]. Our study delves into the knowledge, attitudes, and practices of community pharmacists in Lahore regarding antibiotic dispensing without prescription, aiming to shed light on a significant issue in a region where the role of community pharmacists is not fully established.

## Materials and methods

### Study design

A descriptive, cross-sectional study was conducted in community pharmacy settings of Lahore, Punjab, Pakistan, to explore the knowledge, attitude, and practices of community pharmacists (CPs) about dispensing antibiotics without prescription (DAwP).

### Study settings

Lahore is a metropolitan city in Pakistan with an approximate population of 15,245,000 [23]. The community pharmacy sector of this region is well-established and has both independent and chain pharmacies. An average of 3,618 community & retail outlets (chain pharmacies = 113, independent pharmacies = 264, and retail drug outlets or medical stores = 3,241) are currently working in Lahore to cater to the needs of the growing population [24].

### Study population

The study population targeted for data collection includes all those community pharmacists (CPs) registered with the Punjab Pharmacy Council (PPC) and practising at community pharmacies in Lahore for a minimum of 8 hours per day. Moreover, only those pharmacies that employed full-time licensed pharmacists in Lahore were included.

**Inclusion & exclusion criteria.** This study encompasses registered pharmacists actively employed in community and retail settings as community pharmacists, holding a valid license. Pharmacists classified as industrial pharmacists and those involved in non-community pharmacy sectors are not considered for inclusion. Furthermore, pharmacy technicians, non-registered pharmacists, and registered pharmacists outside Lahore are excluded from the study.

### Sample size

We calculated a sample size of 348 using the Raosoft sample size calculator online, applying a 5% margin of error and a 95% confidence level, considering a population size of 3618. Our study focused on diverse community and retail pharmacy setups. We engaged with 357 licensed community pharmacists (CPs) in Lahore, Pakistan, encompassing both chain

pharmacies (113) and independent pharmacy outlets (244). To enhance the robustness of our study, we collected more data than the calculated sample size, thereby reducing the likelihood of errors in our findings.

## Questionnaire development

The investigational team adopted and modified a questionnaire from the previously published studies [15, 18, 25, 26]. The expert opinions of senior researchers and CPs were also considered to validate this tool. SPSS tool (version 26) was used to calculate reliability coefficients. Cronbach's alpha measured internal consistency, while reproducibility was evaluated using intraclass correlation for each item in the scales, with acceptable values ≥0.6. The Cronbach's alpha value was 0.84 for the knowledge and attitude section and 0.83 for the practices section, demonstrating excellent reliability. A pilot study was undertaken on a small group of 15 participants for pretesting the study instrument. The recommended changes were adjusted in the questionnaire. The final version of the questionnaire had 27 items. These questions were subdivided into 4 main sections. The first section consisted of 7 questions about the demographic details of the CPs. The second section had 11 questions regarding the knowledge and attitude of pharmacists towards DAwP. Each question was rated on a five-point Likert scale (Strongly disagree = 1, Disagree = 2, Neutral = 3, Agree = 4, Strongly agree = 5). The third section comprised four questions that explain why pharmacists are compelled to dispense without prescription. The final section had six questions about the practices of DAwP.

## Data collection

According to the study's objectives, data were collected between March 1, 2023, and March 31, 2023. The data was collected by trained pharmacy students under the supervision of the principal investigator. Investigators visited the pharmacies and invited the CPs to fill out the questionnaire after explaining the purpose of the study. If more than one pharmacist worked at a community pharmacy, both were requested to complete the questionnaire. The CPs either completed the questionnaire instantly or asked the investigator to collect the questionnaire at a mutually agreed time.

## Statistical analysis

Data were analyzed using Statistical Package for Social Sciences (IBM SPSS Statistics for Windows, version 26, Armonk, NY: IBM Corp.). Descriptive statistics, such as frequencies, percentages, median, and interquartile range (IQR), were used to present the data. Kolmogorov-Smirnov and Shapiro-Wilks tests were carried out to test the normality of the data. Furthermore, logistic regression analysis was performed to determine the factors associated with the pharmacists' DAwP practices. Results were expressed as Odds Ratio (OR) accompanied by 95% Confidence Intervals (95% CI), and a p-value $<0.05$ was used for the statistical significance of differences. Outcomes regarding DAwP practices of the pharmacists were dichotomized as "Good" versus "Poor." Scores of ≥14 were considered "Good," whereas scores $<14$ were regarded as "Poor".

## Ethical consideration

The ethical approval was obtained from the Human Research Ethics Committee (HREC) at Cancer Care Hospital and Research Centre (Reference: 29-2023/HREC, January 29, 2023). Before conducting the study, the study protocols were explained to all the participants, and their written consent was also obtained.

**Table 1. Characteristics of the respondents.**

| Variables | | n (%) |
|---|---|---|
| Age (years) | ≤30 | 251 (69.9) |
| | 31–40 | 88 (24.5) |
| | >40 | 20 (5.6) |
| Gender | Male | 159 (44.3) |
| | Female | 200 (55.7) |
| Job-status | Owner | 27 (7.5) |
| | Manager | 118 (32.9) |
| | Staff pharmacist | 214 (59.6) |
| Community practice experience (years) | <3 | 170 (47.4) |
| | 3–5 | 134 (37.3) |
| | >5 | 55 (15.3) |
| Medicines dispensed per day | ≤100 | 63 (17.5) |
| | 101–200 | 205 (57.1) |
| | 201–300 | 59 (16.4) |
| | >300 | 32 (8.9) |
| Antibiotics dispensed per day | ≤25 | 205 (57.1) |
| | 26–50 | 122 (34.0) |
| | >50 | 32 (8.9) |

## Results

393 CPs were approached, and 359 consented participants (response rate = 91.4%) were included according to inclusion & exclusion criteria. Out of which, more than half were females (n = 200, 55.7%) and more than two-thirds were ≤30 years of age (n = 251, 69.9%). 59.6% (n = 214) were staff pharmacists and 47.4% (n = 170) had work experience of <3 years. Each day, more than half of the participants dispense 101–200 medicines (n = 205, 57.1%) and ≤25 antibiotics (Table 1).

### Knowledge and attitudes towards DAwP

All 359 participants responded to eleven questions regarding their knowledge and attitudes towards DAwP. Out of a maximum score, *i.e.*, 5 (100%) for the knowledge and attitude towards DAwP, the respondents obtained a median score of 4.0 (IQR = 1), demonstrating good understanding and a fair, positive attitude towards DAwP.

Majority of the respondents (n = 298, 83.0%) agreed (Strongly agreed (SA) + Agreed (A)) with the statement *"DAwP is contributing to the development of antimicrobial resistance"* (Median = 4.0, IQR = 1). Similarly, most of the respondents (n = 294, 81.9%) agreed (Strongly agreed (SA) + Agreed (A)) with the statement *"Antibiotic resistance has become a public health issue"* (Median = 4.0, IQR = 1). For details, please refer to Table 2.

### Reasons for practicing DAwP

The most common reasons attributable to DAwP were found to be the unwillingness of patients to visit physicians in non-serious infections (n = 270, 75.2%), good knowledge of pharmacists about the use of antibiotics (n = 183, 51.0%), and lack of awareness about rules and regulations against DAwP (n = 163, 45.4%). The majority of the antibiotics dispensed without prescription were from the class cephalosporin (n = 260, 72.4%), followed by penicillin (n = 254, 70.8%) and tetracyclines (n = 170, 47.4%). Most of the antibiotics were dispensed in

**Table 2. Knowledge and attitudes towards dispensing antibiotics without prescription.**

| Variables | Strongly Disagree n (%) | Disagree n (%) | Neutral n (%) | Agree n (%) | Strongly Agree n (%) | Median (IQR) |
|---|---|---|---|---|---|---|
| DAwP is a legal practice in Pakistan* | 63 (17.5) | 161 (44.8) | 72 (20.1) | 59 (16.4) | 4 (1.1) | 4.0 (1) |
| DAwP is common among community pharmacists in Pakistan | 8 (2.2) | 56 (15.6) | 70 (19.5) | 203 (56.5) | 22 (6.1) | 4.0 (1) |
| Do you think there is any problem if you dispense medication without a prescription | 8 (2.2) | 46 (12.8) | 41 (11.4) | 210 (58.5) | 54 (15.0) | 4.0 (1) |
| DAwP is contributing to the development of antimicrobial resistance | 4 (1.1) | 12 (3.3) | 45 (12.5) | 157 (43.7) | 141 (39.3) | 4.0 (1) |
| Antibiotic resistance has become a public health issue | 0 (0.0) | 29 (8.1) | 36 (10.0) | 157 (43.7) | 137 (38.2) | 4.0 (1) |
| DAwP is contributing to the inappropriate use of antibiotics by patients | 15 (4.2) | 11 (3.1) | 68 (18.9) | 196 (54.6) | 69 (19.2) | 4.0 (1) |
| Pharmacists can be penalized for DAwP | 52 (14.5) | 116 (32.3) | 79 (22.0) | 101 (28.1) | 11 (3.1) | 3.0 (2) |
| Pharmacists should stop DAwP | 8 (2.2) | 61 (17.0) | 57 (15.9) | 175 (48.7) | 58 (16.2) | 4.0 (1) |
| I encourage patients to consult the physician and get a prescription | 60 (16.7) | 143 (39.8) | 47 (13.1) | 60 (16.7) | 49 (13.6) | 4.0 (2) |
| When patients feel that they need an antibiotic, if not dispensed, they will try to obtain it from another pharmacy | 31 (8.6) | 58 (16.2) | 49 (13.6) | 147 (40.9) | 74 (20.6) | 2.0 (1) |
| Refusing DAwP will negatively affect sales and profits* | 12 (3.3) | 78 (21.7) | 111 (30.9) | 118 (32.9) | 40 (11.1) | 3.0 (2) |
| **Overall** | | | | | | **4.0 (1)** |

IQR = Interquartile range;

*Negative statement.

*Note*: Knowledge and attitude were assessed by giving 1 to disagree Strongly, 2 to Disagree, 3 to Neutral, 4 to Agree, and 5 to agree Strongly. Questions 1 and 11 were assessed by giving 5 to Strongly disagree, 4 to Disagree, 3 to Neutral, 2 to Agree, and 1 to agree Strongly.

oral dosage form (n = 238, 66.3%). Also, cold and flu (n = 223, 62.1%) and diarrhea (n = 182, 50.7%) were the most common conditions for which antibiotics were dispensed without prescription (Table 3).

## Practices towards DAwP

All 357 participants responded to six questions regarding their practices towards DAwP. Out of a maximum score, *i.e.*, 3 (100%) for the practices of CPs towards DAwP, the respondents obtained a median score of 2 (IQR = 0.5), demonstrating good practices towards DAwP.

79.1% of the CPs (n = 284) never/sometimes warn patients about the potential side effects of medicines when dispensing antibiotics without prescription (Median = 2.0, IQR = 1). Similarly, just 51.0% of the CPs (n = 183) continually educate patients about the importance of adherence and completing the entire course of antibiotics when dispensing antibiotics without a prescription (Median = 3.0, IQR = 1) (Table 4).

The logistic regression analysis results examined the association between DAwP practices of the CPs and the independent variables (*i.e.*, age, gender, job status, experience, antibiotics dispensed daily, and medicines administered daily). Results revealed that CPs aged 31–40 years (OR = 0.568, 95% CI = 0.348–0.927, p-value = 0.024) were 0.568 times significantly less likely to be associated with poor practices of DAwP compared to those aged ≤30 years. CPs whose job status was 'Manager' (OR = 4.222, 95%CI = 2.542–7.011, *p-value* = <0.001) were 4.222 times more likely to be associated with poor DAwP practices than 'Staff pharmacists.' While examining the association with experience, CPs with 3–5 years of experience (OR = 2.241, 95%CI = 1.183–4.243, *p-value* = 0.013) were 2.241 times more likely to be associated with poor practices of DAwP than those with >5 years of experience. Furthermore, CPs who dispense ≤25 antibiotics per day (OR = 12.375, 95%CI = 5.177–29.583, *p-value* = <0.001) were 12.375 times more likely to be associated with poor practices of DAwP as compared to those who dispense >50 antibiotics per day. What is more, CPs who dispense ≤100 medicines

**Table 3. Classes of antibiotics and reasons for practising DawP.**

| Variables | | n (%) |
|---|---|---|
| Reasons for DAwP | Pharmacists have good knowledge about antibiotic use | 183 (51.0) |
| | Patients do not want to see a doctor unless the infection is serious | 270 (75.2) |
| | Increased sales and profit pressure by the owner | 109 (30.4) |
| | Patients cannot afford to consult a physician | 146 (40.7) |
| | Fear of losing a client/patient | 100 (27.9) |
| | Lack of awareness about rules and regulations against DAwP | 163 (45.4) |
| Commonly dispensed antibiotic classes. | Penicillin | 254 (70.8) |
| | Cephalosporin | 260 (72.4) |
| | Macrolides | 147 (40.9) |
| | Quinolones + Flouroquinolones | 159 (44.3) |
| | Tetracyclines | 170 (47.4) |
| | Aminoglycosides | 166 (46.2) |
| | Sulfonamides | 117 (32.6) |
| Common antibiotic dosage forms dispensed without a prescription | Oral | 238 (66.3) |
| | Eye drops | 211 (58.8) |
| | Ear drops | 90 (25.1) |
| | Topical | 171 (47.6) |
| | Injectable | 7 (1.9) |
| Medical conditions for which antibiotics are dispensed without a prescription | Rhinitis | 87 (24.3) |
| | Diarrhea | 182 (50.7) |
| | Cold and flue | 223 (62.1) |
| | Toothache | 76 (21.2) |
| | Earache | 100 (27.9) |
| | Other (e.g., eye infection, wound, UTIs) | 170 (47.4) |

per day (OR = 8.812, 95%CI = 3.305–23.500, *p-value* = <0.001) were 8.812 times and those who dispense 101–200 drugs per day (OR = 4.151, 95%CI = 1.780–9.682, *p-value* = <0.001) were 4.151 times more likely to be associated with poor practices of DAwP as compared to those who dispense >300 medicines per day (Table 5).

**Table 4. DAwP practices of the pharmacists.**

| Variables | Never n (%) | Sometimes n (%) | Always n (%) | Median (IQR) |
|---|---|---|---|---|
| When dispensing antibiotics, I ask patients about drug allergies | 80 (22.3) | 152 (42.3) | 127 (35.4) | 2.0 (1) |
| When dispensing antibiotics without a prescription, I ask patients if they have any kidney problem | 87 (24.2) | 179 (49.9) | 93 (25.9) | 2.0 (1) |
| When dispensing antibiotics without a prescription, I warn patients about the potential side effects of medicines | 93 (25.9) | 191 (53.2) | 75 (20.9) | 2.0 (1) |
| When dispensing antibiotics without a prescription, I educate patients about the importance of adherence and completing the entire course of antibiotics | 78 (21.7) | 98 (27.3) | 183 (51.0) | 3.0 (1) |
| When dispensing antibiotics without a prescription, if they are taking any other medication for the same complaint | 54 (15.0) | 153 (42.6) | 152 (42.3) | 2.0 (1) |
| I don't dispense antibiotics without a prescription for children | 93 (25.9) | 117 (32.6) | 149 (41.5) | 2.0 (2) |
| **Overall** | | | | **2.0 (0.5)** |

IQR = Interquartile range; *Note*: Practices were assessed by giving 1 to Never, 2 to Sometimes, and 3 to Always.

**Table 5. Logistic regression analysis of factors associated with pharmacists' DAwP practices.**

| Characteristics | | DAwP Practices | | OR | 95%Cl | P-values |
|---|---|---|---|---|---|---|
| | | Good | Poor | | | |
| Age (years) | ≤30 | 99 (27.6) | 152 (42.3) | 1.0 | - | - |
| | 31–40 | 47 (13.1) | 41 (11.4) | 0.568 | 0.348–0.927 | **0.024** |
| | >41 | 12 (3.3) | 8 (2.2) | 0.434 | 0.171–1.100 | 0.079 |
| Gender | Male | 77 (21.4) | 82 (22.8) | 0.725 | 0.476–1.103 | 0.133 |
| | Female | 81 (22.6) | 119 (33.1) | 1.0 | - | - |
| Job-status | Owner | 12 (3.3) | 15 (4.2) | 1.566 | 0.700–3.504 | 0.275 |
| | Manager | 27 (7.5) | 91 (25.3) | 4.222 | 2.542–7.011 | **<0.001** |
| | Staff pharmacist | 119 (33.1) | 95 (26.5) | 1.0 | - | - |
| Experience (years) | <3 | 78 (217) | 92 (25.6) | 1.524 | 0.826–2.811 | 0.178 |
| | 3–5 | 49 (13.6) | 85 (23.7) | 2.241 | 1.183–4.243 | **0.013** |
| | >5 | 31 (8.6) | 24 (6.7) | 1.0 | - | - |
| Antibiotics dispensed per day | ≤25 | 40 (11.1) | 165 (46.0) | 12.375 | 5.177–29.583 | **<0.001** |
| | 26–50 | 94 (26.2) | 28 (7.8) | 0.894 | 0.362–2.208 | 0.807 |
| | >50 | 24 (6.7) | 8 (2.2) | 1.0 | - | - |
| Medicines dispensed per day | ≤100 | 16 (4.5) | 47 (13.1) | 8.812 | 3.305–23.500 | **<0.001** |
| | 101–200 | 86 (24.0) | 119 (33.1) | 4.151 | 1.780–9.682 | **<0.001** |
| | 201–300 | 32 (8.9) | 27 (7.5) | 2.531 | 0.979–6.545 | 0.055 |
| | >300 | 24 (6.7) | 8 (2.2) | 1.0 | - | - |

## Discussion

The present study is designed to evaluate the knowledge, attitudes, and practices of community pharmacists (CPs) concerning Drug Adherence with Prescription (DAwP). The impetus behind this research is to provide healthcare policymakers with crucial insights into the perspectives and existing behaviours of CPs. Understanding these elements is essential for shaping effective strategies to improve patient-centred pharmaceutical services in community pharmacies. Moreover, this study is particularly significant in the context of antimicrobial stewardship. By assessing how CPs adhere to and advise on prescriptions, especially regarding antibiotics, the study can contribute to broader efforts to promote responsible antimicrobial use and combat the growing challenge of antibiotic resistance.

Findings revealed that the CPs had good knowledge and a positive attitude towards DAwP. These results are comparable with the study conducted in Albania, in which more than half of the CPs had adequate knowledge about DAwP [27]. A contrast was found in a study conducted in Saudi Arabia, where 50–75% of the CPs had poor knowledge of DAwP [28]. Although worldwide pharmacy curricula have been designed to allow pharmacists to learn the basic concept of rational dispensing practices, the difference in training may cause a divergence in the knowledge of CPs [29]. Most CPs agreed with the statement, *"DAwP is contributing to the development of antimicrobial resistance."* Similar results were obtained from a survey-based study conducted by the Netherlands Institute for Health Services Research (NIVEL) in Europe, where 90% of the CPs agreed that DAwP may increase the resistance crisis up to several folds [30]. Studies from different regions of the globe predict the increasing problem of antimicrobial resistance as a negative outcome of DAwP [25, 26, 31–33]. Also, like the previously published studies [25, 34, 35], most CPs confirmed that *"Antibiotic resistance has become a public health issue."* Thus, the positive attitude towards DAwP is attributable to the fact that CPs don't hesitate to dispense drugs without prescription because community pharmacies are private setups where priority is given to revenue instead of patient safety in Pakistan.

The present study reported that the most common reasons attributable to DAwP were the unwillingness of patients to visit physicians for non-serious infections, good knowledge of CPs about the use of antibiotics, and lack of awareness about rules and regulations against DAwP. Likewise, a study reported a shortage of understanding among CPs regarding the laws of the rational dispensing of antibiotics [36]. The evidence further suggested that in many evolving economies like Pakistan, patients don't consider it necessary to consult physicians and tend to self-treat their minor ailments with antibiotics because of the influence of friends and family members, accessibility of medicines without prescription, unavailability of time, money and other resources; and traditional beliefs [37, 38]. Also, due to financial constraints, many patients purchase medicines according to their budget but not per the prescription. Thus, they buy half rather than the entire course of treatment [39].

Most antibiotics dispensed without prescription were from the class cephalosporin, followed by penicillin and tetracyclines. A qualitative study in Ethiopia revealed that penicillins and cephalosporins were the most frequently sold antibiotics without prescription [40]. Several studies showed that nearly 50% of antibiotics are dispensed without prescription in various regions, including Pakistan, because of poor regulation and lack of guidelines for rational dispensing practices [11, 41–44]. It was also found that most of the antibiotics were dispensed in oral dosage form. This finding aligns with a cross-sectional study where 86.8% of the antibiotics were allocated in oral dosage form because these were easy to administer [1]. Also, like the previously published studies [45–47], cold, flu, and diarrhea were the most common conditions for which antibiotics were dispensed without a prescription. This might be because patients consider these ailments minor and are reluctant to consult physicians [48].

The present study further revealed that community pharmacists (CPs) wrongly claimed good practices towards DAwP. However, it was noted that a majority of CPs either never or only sometimes informed patients about the potential side effects of medicines when dispensed with antibiotics without a prescription (DAwP). Similarly, half of the CPs always educate patients about the importance of adherence and completing the entire course of antibiotics when dispensing antibiotics without a prescription. A cross-sectional study conducted in Pakistan evaluated that DAwP was practised by 34% of the CPs [22]. Another multi-center survey-based study conducted in China revealed that antibiotics were dispensed without a prescription in 55.9% of the pharmacies, and proper counselling was given in 17.5% of the community pharmacies [13]. This could be attributed to the underdevelopment and inadequate implementation of community pharmacy practices in the majority of community

settings in Pakistan. In such contexts, the role of community pharmacists is frequently confined to procuring and dispensing medicines. As a result, these healthcare professionals may not have adequate time to dedicate to patient-oriented services [49, 50].

Logistic regression analysis revealed that age, job status, experience of CPs, and the number of medicines dispensed per day significantly correlated with the practice of DAwP. Findings of the current study showed that the CPs of 31–40 years of age were less likely to be associated with poor practices of DAwP than those of ≤30 years of age. A similar association between CPs' periods and dispensing practices was found in a Spanish study [51]. In Pakistan, antimicrobial stewardship (AMS) training programs are unavailable at undergraduate and postgraduate levels, and there is a shortage of practice-based learning. Therefore, young pharmacists show good DAwP practices. It was also revealed that CPs with job status as 'Manager' and 3–5 years of experience were more likely to be associated with poor DAwP practices than 'Staff pharmacists' and those with>5 years of experience. These findings are comparable with the studies in which CP's work status and job experience were significantly associated with the DAwP practices [51–53]. This might be because, in many developing countries like Pakistan, there is no establishment of continuing medical education (CME) programs by which CPs can keep their knowledge up to date [54]. Furthermore, like the previously published studies [55–57], CPs who dispense fewer antibiotics (≤25 antibiotics) per day were more likely to be associated with poor practices of DAwP as compared to those who dispense a more significant number of antibiotics per day. As the workload of CPs has increased due to emerging urbanization in Pakistan, they can't find ample time to ask patients questions about their prescriptions and thus cannot practice the rational dispensing of antibiotics.

Hence, it is recommended that policymakers develop guidelines for rational dispensing of antibiotics for CPs. There is a need for drug awareness campaigns regarding DAwP-associated health consequences not only for CPs but also for the general public. Also, federal and provincial health ministries should organize CME programs, AMS training sessions, and health awareness seminars, and CPs must actively participate in such campaigns.

## Strength and limitations

The study presents a unique contribution by investigating community pharmacists' knowledge, attitudes, and practices towards DAwP in Lahore, Pakistan. To the authors' knowledge, no prior research has explored this topic within Lahore or any administrative division of the Lahore District, or elsewhere in Pakistan. This original focus strengthens the study's significance and relevance to the local context. This study contributes valuable insights for future researchers, policymakers, and stakeholders.

The current study has few limitations. First, the findings can't be generalized nationwide because this study was conducted briefly. As the condition and size of community pharmacies and the job experience of CPs are not similar across the country, results are less likely to be identical for other community pharmacy settings. Second, the biases of self-administered questionnaires arise due to differences in accuracy or completeness of recollections of participants and to under or over-reporting of knowledge, attitude, and practices of CPs towards DAwP. Furthermore, the investigators could not find differences in the actual practice of CPs from their claim because no follow-up was performed, and all the CPs were approached only once during the data collection.

## Conclusion

The present study concludes that community pharmacists (CPs) generally possess good knowledge and poor attitudes towards DAwP. However, there are notable gaps in their

practices, particularly regarding patient education on the potential side effects of DAwP. Factors such as age, job status, experience of CPs, and the number of medicines dispensed per day were identified as significant determinants of DAwP. These findings underscore the need for targeted interventions to improve CPs' practices and enhance patient safety. Healthcare policymakers should be made aware of these findings to inform the development of effective strategies to address antimicrobial resistance crises associated with DAwP. Additionally, there is a call for initiating practice-based training programs at both federal and provincial levels in Pakistan to enhance CPs' competencies in antimicrobial stewardship and patient education.

## Acknowledgments

I express my sincere gratitude to the Cancer Care Hospital & Research Centre for their invaluable support and collaboration in conducting this research. Their commitment to advancing healthcare in Lahore, Pakistan, has been instrumental in our investigation of antibiotic adherence in community pharmacies.

## Author Contributions

**Conceptualization:** Muhammad Nabeel, Khubaib Ali.

**Data curation:** Muhammad Nabeel, Khubaib Ali.

**Formal analysis:** Muhammad Nabeel, Khubaib Ali.

**Investigation:** Muhammad Nabeel, Muhammad Rehan Sarwar.

**Methodology:** Muhammad Nabeel.

**Resources:** Muhammad Nabeel.

**Software:** Muhammad Nabeel, Khubaib Ali.

**Supervision:** Muhammad Rehan Sarwar, Imran Waheed.

**Validation:** Muhammad Rehan Sarwar.

**Writing – original draft:** Muhammad Nabeel, Khubaib Ali.

**Writing – review & editing:** Muhammad Nabeel, Khubaib Ali, Muhammad Rehan Sarwar.

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
