## [Decision Letter · Decision Letter 0]

2 Nov 2023

PONE-D-23-29837Prescription Precision: Investigating Antibiotic Adherence in Community Pharmacies - A Cross-Sectional Study from Lahore, PakistanPLOS ONE

Dear Dr. Nabeel,

Thank you for submitting your manuscript to PLOS ONE. After careful consideration, we feel that it has merit but does not fully meet PLOS ONE’s publication criteria as it currently stands. Therefore, we invite you to submit a revised version of the manuscript that addresses the points raised during the review process.

We look forward to receiving your revised manuscript.

Kind regards,

Naeem Mubarak, PhD

Academic Editor

PLOS ONE

A clean copy of the edited manuscript (uploaded as the new *manuscript* file)”.

Reviewers' comments:

Reviewer's Responses to Questions

**Comments to the Author**

1. Is the manuscript technically sound, and do the data support the conclusions?

Reviewer #1: Yes

Reviewer #2: Yes

2. Has the statistical analysis been performed appropriately and rigorously? 

Reviewer #1: Yes

Reviewer #2: Yes

3. Have the authors made all data underlying the findings in their manuscript fully available?

Reviewer #1: Yes

Reviewer #2: Yes

4. Is the manuscript presented in an intelligible fashion and written in standard English?

Reviewer #1: No

Reviewer #2: Yes

5. Review Comments to the Author

Reviewer #1: Comment 1: In the Methodology Section, at line 133, the abbreviations "CPs" and "DAwP" are utilized. I would recommend expanding these abbreviations to enhance reader comprehension.

Comment 2: Within the Methodology Section, it would be beneficial to provide further clarification regarding the sample size. Specifically, it would be helpful to explain how the sample size was determined and indicate the source of the calculation.

Comment 3: In the Questionnaire Development Section, at Line 152, additional clarification is needed regarding the version of SPSS software used. The abstract mentions the use of SPSS version 26 for statistical analysis, whereas here, it is stated as the 21st version. This discrepancy should be addressed.

Comment 4: Throughout the entire manuscript, I recommend a thorough review to identify and rectify any grammatical mistakes, spelling errors, and typographical errors.

Comment 5: Additionally, please check the formatting of the manuscript to ensure it adheres to standard guidelines. Remove any extraneous lines or spacing issues, such as those found at lines 256, 257, 377, and 529, to maintain a professional appearance.

Reviewer #2: Thank you for providing me the opportunity to review this manuscript. This survey based study focuses on determining the knowledge, attitude and practices of community pharmacists in Pakistan on dispensing antibiotics without prescription. The study merits contributing betterment in antibiotic dispensing practices and containing antibiotic resistance in Pakistan. However, after critical observation, some areas need further consideration before the manuscript deems suitable for publication. Following are the recommendations for the authors to consider

Title:

The title of the manuscript needs revision as the term “prescription precision” and “antibiotic adherence” does not clearly align with the objectives of the study.

Background:

It is suggested to use antibiotic resistance instead “antimicrobial resistance” when especially denoting resistance from bacterial infections. The terminology should be coherent throughout.

Please provide more details on how irrational antibiotic practices have contributed toward antibiotic resistance in Pakistan and what innovative findings your study presents that add difference to the already existing literature.

The references used in introduction section need updation. For instance, reference # 2,3,4,5 and 6. To provide a more holistic background to antibiotic resistance in Pakistan and factors contributing it, the following study may provide a more solid rationale to the objectives of your study. (This is optional and should be taken as a suggestion for the improvement of the manuscript)

1. PMCID: PMC8388777

2. PMCID: PMC7602895

Methodology:

The manuscript provides no details on the total population size of CPs used and how the sample size is calculated? Are medical stores included in the calculation of sample size? Hence, mention the inclusion/exclusion criteria in detail.

Also provide more clarity on the development and validity of your questionnaire. How many items (added/deleted/modified) from the idea generation phase

The figures provided on the total community pharmacies in Pakistan are taken from 2017 study. Have the figures remained same until now? If not, would they affect the calculation of sample size? Please justify.

For lines; 171-173: For the better understanding, please provide details on how frequent visits were made to achieve the desired response rate

Discussion:

In discussion section, whenever we discuss antibiotic resistance, it is crucial to discuss antimicrobial stewardship as well. However, no such mentioning can be seen.

The references used throughout the discussion section need updation. For instance, in ref 28 and 52, when discussing about how undergraduate training may affect the practices of CPs on antibiotic dispensing, the following updated study can be a relevant and appropriate reference to your study. (This is optional and should be taken as a suggestion for the improvement of the manuscript)

PMCID: PMC8532898

6. PLOS authors have the option to publish the peer review history of their article (what does this mean?). If published, this will include your full peer review and any attached files.

Reviewer #1: No

Reviewer #2: No

---

## [Author Response · Author response to Decision Letter 0]

14 Dec 2023

I would like to express my sincere gratitude to the reviewers for their time and effort in evaluating our manuscript and for providing constructive feedback. All highlighted issues have been addressed comprehensively, which I trust will facilitate a more expedient publication process. Thank you.

---

## [Decision Letter · Decision Letter 1]

9 Jan 2024

PONE-D-23-29837R1Assessment of Knowledge, Attitudes, and Practices among Community Pharmacists in Lahore Regarding Antibiotic Dispensing without Prescription: A Cross-Sectional StudyPLOS ONE

Dear Dr. Muhammad Nabeel,

Thank you for submitting your manuscript to PLOS ONE. After careful consideration, we feel that it has merit but does not fully meet PLOS ONE’s publication criteria as it currently stands. Therefore, we invite you to submit a revised version of the manuscript that addresses the points raised during the review process.

We look forward to receiving your revised manuscript.

Kind regards,

Naeem Mubarak, PhD

Academic Editor

PLOS ONE

Journal Requirements:

Additional Editor Comments (if provided):

The concerns presented by reviewer 3 are necessary to address to further improve the quality and essence of the manuscript.

Reviewers' comments:

Reviewer's Responses to Questions

**Comments to the Author**

1. If the authors have adequately addressed your comments raised in a previous round of review and you feel that this manuscript is now acceptable for publication, you may indicate that here to bypass the “Comments to the Author” section, enter your conflict of interest statement in the “Confidential to Editor” section, and submit your "Accept" recommendation.

Reviewer #2: All comments have been addressed

Reviewer #3: (No Response)

2. Is the manuscript technically sound, and do the data support the conclusions?

Reviewer #2: Yes

Reviewer #3: Yes

3. Has the statistical analysis been performed appropriately and rigorously? 

Reviewer #2: Yes

Reviewer #3: Yes

4. Have the authors made all data underlying the findings in their manuscript fully available?

Reviewer #2: Yes

Reviewer #3: Yes

5. Is the manuscript presented in an intelligible fashion and written in standard English?

Reviewer #2: Yes

Reviewer #3: Yes

6. Review Comments to the Author

Reviewer #2: The authors have addressed all the previous concerns and suggestions. I have no further comments to add.

Reviewer #3: Introduction

Comment 1

Page 6, line 129-130: Consider providing more specific details regarding the aim of the present study that will give a distinction from the previous publications within the same area and scope.

Discussion

Comment 2

Page 13, line 250: ‘’…were significantly 43.2% less likely…’’

Consider mentioning the times rather than the percentage.

Comment 3

Page 17, line 315-316: ‘’ The present study further revealed that the CPs had good practices towards DAwP. Most CPs never or sometimes warn the patients about the potential side effects of medicines when dispensed with antibiotics without a prescription.’’

Consider revising and rephrasing these sentences as there is an obvious conflict between the good practices and never or sometimes providing warning to the patients about the potential side effects!

Comment 4

Page 17, line 323: ‘’ This might be because the pharmacy profession is in its infancy in Pakistan.’’

What are the authors trying to say in this sentence. Is it correct that the pharmacy profession is in infancy time in this country!

Strength and limitations

Comment 5

Page 19, line 351: ‘’ no previously published data is available in any South Asian’’

Authors are required not to generalize their findings to the surrounding geographical area as the main scope of the study was related to Pakistan.

Consider removing this sentence.

Comment 6

Page 19, line 355: ‘’but also worldwide’’.

Similarly, authors are required not to generalize their findings as the main scope of the study was related to Pakistan. Consider removing this sentence.

Conclusion

Comment 7

Page 17, line 365: ‘’The present study concluded that CPs had good knowledge and positive attitudes towards DAwP’.

Consider rephrasing this sentence. Since, based on the current study findings, there is a relatively good knowledge but poor attitudes towards DAwP’’. Consider mentioning this point as a part of the conclusion within the conclusion of both the abstract and main text.

References

Comment 8: Consider revising and correcting the following list of references.

Reference (2): This reference is outdated. Consider providing a more recent one.

Reference (3): Provide the website link.

Reference (6): This reference is outdated. Consider providing a more recent one.

Reference (8): Provide the website link.

Reference (9): This reference is outdated. Consider providing a more recent one, including the one

doi: 10.3855/jidc.13066.

Reference (10): This reference is unrelated to the sentence reported. Consider correction.

Reference (23): Provide the website link.

Reference (28): Consider removing the data from this reference and replacing a more appropriate recent one with data within its reference.

Reference (30): Provide the website link.

Reference (36): This reference is unrelated to the sentence reported. Consider correction.

7. PLOS authors have the option to publish the peer review history of their article (what does this mean?). If published, this will include your full peer review and any attached files.

Reviewer #2: No

Reviewer #3: **Yes: **Dr. Anmar AL-TAIE

---

## [Author Response · Author response to Decision Letter 1]

10 Feb 2024

Thank you for reviewing my manuscript and bringing the mistakes to my attention. I have made the necessary revisions and would like to resubmit it. I am confident that this revised version meets the standards of PLOS ONE and hope for its timely publication.

---

## [Decision Letter · Decision Letter 2]

29 Feb 2024

PONE-D-23-29837R2Assessment of Knowledge, Attitudes, and Practices among Community Pharmacists in Lahore Regarding Antibiotic Dispensing without Prescription: A Cross-Sectional StudyPLOS ONE

Dear Dr. Nabeel,

Thank you for submitting your manuscript to PLOS ONE. After careful consideration, we feel that it has merit but does not fully meet PLOS ONE’s publication criteria as it currently stands. Therefore, we invite you to submit a revised version of the manuscript that addresses the points raised during the review process.

Kind regards,

Naeem Mubarak, PhD

Academic Editor

PLOS ONE

Journal Requirements:

Additional Editor Comments :

The manuscript may be accepted for publication after minor revisions

Reviewers' comments:

Reviewer's Responses to Questions

**Comments to the Author**

1. If the authors have adequately addressed your comments raised in a previous round of review and you feel that this manuscript is now acceptable for publication, you may indicate that here to bypass the “Comments to the Author” section, enter your conflict of interest statement in the “Confidential to Editor” section, and submit your "Accept" recommendation.

Reviewer #2: All comments have been addressed

Reviewer #3: (No Response)

2. Is the manuscript technically sound, and do the data support the conclusions?

Reviewer #2: Yes

Reviewer #3: Yes

3. Has the statistical analysis been performed appropriately and rigorously? 

Reviewer #2: Yes

Reviewer #3: Yes

4. Have the authors made all data underlying the findings in their manuscript fully available?

Reviewer #2: Yes

Reviewer #3: Yes

5. Is the manuscript presented in an intelligible fashion and written in standard English?

Reviewer #2: Yes

Reviewer #3: Yes

6. Review Comments to the Author

Reviewer #2: The authors have addressed all my concerns. I have no suggestions to add further. Best of luck with your publication.

Reviewer #3: There are still remaining critics that did not get into consideration and corrected by the authors, as below:

Introduction:

Page 5: line 15-17: “Studies depict that such irrational dispensing practices are also commonly found in various Gulf countries like Saudi Arabia (14, 15), Egypt (16), Jordan (17), and Syria (18)”.

The sentence is incorrect and needs to be observed and rephrased since some of the countries are not a part of the Gulf, such as Egypt, Jordan and Syria.

The authors can mention Gulf countries like Saudi Arabia with the citations (14, 15). Also it is better to consider another example of the Gulf countries and Middle East like Iraq with the citation (doi: 10.3855/jidc.13066). Then it can mention other countries, like Egypt, Jordan and Syria.

References

Reference (3): Invalid link. Provide the correct website link.

Reference (28): Incorrect. Consider corrected one.

Reference (30): Invalid link. Provide the correct website link.

7. PLOS authors have the option to publish the peer review history of their article (what does this mean?). If published, this will include your full peer review and any attached files.

Reviewer #2: No

Reviewer #3: No

---

## [Author Response · Author response to Decision Letter 2]

11 Apr 2024

The reviewer's minor concerns have been properly addressed, and I am optimistic that the article will now be accepted for publication soon. Thank you.

---

## [Decision Letter · Decision Letter 3]

13 May 2024

Assessment of Knowledge, Attitudes, and Practices among Community Pharmacists in Lahore Regarding Antibiotic Dispensing without Prescription: A Cross-Sectional Study

PONE-D-23-29837R3

Dear Dr. Muhammad Nabeel,

We’re pleased to inform you that your manuscript has been judged scientifically suitable for publication and will be formally accepted for publication once it meets all outstanding technical requirements.

Kind regards,

Naeem Mubarak, PhD

Academic Editor

PLOS ONE

Additional Editor Comments (optional):

The manuscript has incorporated all the necessary changes. No further revisions are required. Best of luck with your publication.

Reviewers' comments:

Reviewer's Responses to Questions

**Comments to the Author**

1. If the authors have adequately addressed your comments raised in a previous round of review and you feel that this manuscript is now acceptable for publication, you may indicate that here to bypass the “Comments to the Author” section, enter your conflict of interest statement in the “Confidential to Editor” section, and submit your "Accept" recommendation.

Reviewer #3: All comments have been addressed

Reviewer #4: All comments have been addressed

2. Is the manuscript technically sound, and do the data support the conclusions?

Reviewer #3: Yes

Reviewer #4: Yes

3. Has the statistical analysis been performed appropriately and rigorously? 

Reviewer #3: Yes

Reviewer #4: Yes

4. Have the authors made all data underlying the findings in their manuscript fully available?

Reviewer #3: Yes

Reviewer #4: Yes

5. Is the manuscript presented in an intelligible fashion and written in standard English?

Reviewer #3: Yes

Reviewer #4: Yes

6. Review Comments to the Author

Reviewer #3: (No Response)

Reviewer #4: The manuscript requires no further changes. All necessary changes have been included. Best of luck with your publication

7. PLOS authors have the option to publish the peer review history of their article (what does this mean?). If published, this will include your full peer review and any attached files.

Reviewer #3: No

Reviewer #4: No

---

## [Editor Report · Acceptance letter]

4 Jun 2024

PONE-D-23-29837R3 

PLOS ONE

Dear Dr. Nabeel, 

I'm pleased to inform you that your manuscript has been deemed suitable for publication in PLOS ONE. Congratulations! Your manuscript is now being handed over to our production team.

Kind regards, 

on behalf of

Dr Naeem Mubarak 

Academic Editor

PLOS ONE